# Brief communication: *Ad hoc* estimation of glacier contributions to sea-level rise from latest glaciological observations

Michael Zemp[1], Matthias Huss[2,3,4], Nicolas Eckert[5], Emmanuel Thibert[5], Frank Paul[1], Samuel U. Nussbaumer[1,3], Isabelle Gärtner-Roer[1]

[1]Department of Geography, University of Zurich, Zurich, 8057, Switzerland
[2]Laboratory of Hydraulics, Hydrology and Glaciology (VAW), ETH Zurich, Zurich, 8093, Switzerland
[3]Department of Geosciences, University of Fribourg, Fribourg, 1700, Switzerland
[4]Swiss Federal Institute for Forest, Snow and Landscape Research (WSL), Birmensdorf, Switzerland
[5]Université Grenoble Alpes, Irstea, UR ETGR, Grenoble, 38402, France

*Correspondence to*: Michael Zemp (michael.zemp@geo.uzh.ch)

**Abstract**

Comprehensive assessments of global glacier mass changes based on a variety of observations and prevailing methodologies have been published at multi-annual intervals. For the years in between, the glaciological method provides annual observations of specific mass changes but is suspected to not be representative at the regional to global scales due to uneven glacier distribution with respect to the full sample. Here, we present a simple approach to estimate and correct for this bias in the glaciological sample and, hence, to provide an *ad hoc* estimate of global glacier mass changes and corresponding sea-level equivalents for the latest years, i.e. about −300 ± 250 Gt in 2016/17 and −500 ± 200 Gt in 2017/18.

## 1 Introduction

Globally, more than 215,000 glaciers – distinct from the Greenland and Antarctic ice sheets – cover an area of about 700,000 km$^2$ (RGI, 2017) with a recently re-estimated total volume of about 160,000 km$^3$ (Farinotti et al., 2019). Glaciers react sensitively to changes in climate (Bojinski et al., 2014) and substantially contribute to regional runoff (e.g., Biemans et al., 2019; Huss and Hock, 2018; Kaser et al., 2010; Pritchard, 2019) and global sea-level change (e.g., Hock et al., 2019; Marzeion et al., 2018). In the context of the Intergovernmental Panel on Climate Change (IPCC) assessment reports, the glaciological community has periodically published observational estimates of glacier contributions to sea-level rise based on coordinated efforts making use of all available data at that time: IPCC TAR (2001): Meier (1984), Trupin et al. (1992); IPCC AR4 (2007): Kaser et al. (2006) based on Cogley (2005), Dyurgerov and Meier (2005), and Ohmura (2004) ; IPCC AR5 (2013): Gardner et al. (2013) mainly based on Cogley (2009). These approaches were challenged by small observational samples covering not more than a few hundred glaciers, with an uneven spatial and temporal distribution (Zemp et al., 2015) and were complemented for IPCC AR5 with estimates from spaceborne altimetry and gravimetry (Gardner et al., 2013). In view of the IPCC Special Report on the Ocean and Cryosphere in a Changing Climate (IPCC, 2019), Zemp et al. (2019) increased the observational sample to more than 19,000 glaciers by combining the results from

glaciological and geodetic (from DEM differencing) methods to assess annual mass changes and corresponding sea-level equivalents from 1961 to 2016. All of these major assessments provided new observational baselines for the comparison with estimates based on other methods such as spaceborne gravimetry or altimetry (e.g., Bolch et al., 2013; Wouters et al., 2019), as well as for modelling studies of future glacier contributions to regional runoff and global sea-level change. In view of the *global stocktake* to assess the collective progress towards achieving the Paris Agreement (cf. UNFCCC, 2016, Article 14), there is an increased demand for regular updates on the state of the climate. However, the approaches underlying these results are unsuitable for providing annual updates on the basis of new glaciological observations acquired each year due to the extensive analysis efforts required and due to generic lack of updates from multi-annual geodetic surveys (from DEM differencing). Here, we present a framework to infer *ad hoc* (i.e., timely but preliminary) estimates of global-scale glacier contributions to sea-level rise from annual updates of glaciological observations. For this purpose, we combine the annual anomaly provided by the glaciological sample (relative to a decadal mean) with the (mean) absolute mass-change rate of a reference dataset (i.e., Zemp et al., 2019) over a common calibration period (from 2006/07 to 2015/16). As a result, we here provide preliminary estimates of regional and global glacier mass changes and related uncertainties for the hydrological years 2016/17 and 2017/18. We also discuss the regional biases of the glaciological sample and conclude with a brief outlook on possible applications and remaining limitations of the glaciological observation network of the World Glacier Monitoring Service (WGMS).

## 2 Data and methods

### 2.1 Regional glacier areas and related change rates

The global distribution of glaciers is taken from the Randolph Glacier Inventory (RGI) version 6.0 (RGI, 2017). This dataset lists 215,547 glaciers covering a total area of 705,739 km$^2$, mainly for survey years between 2000 and 2010. The glaciers in the RGI are grouped into 19 first-order regions, which seems to be appropriate with respect to the spatial correlation distance of glacier mass-balance variability (that is, several hundred kilometres; Cogley and Adams, 1998; Letréguilly and Reynaud, 1990). We consider changes in glacier area over time by using annual change rates for all first-order regions from Zemp et al. (2019, and references therein), based on a data collection from the literature. The regional glacier surface area $S$ for a given year $t_1$ was calculated as:

$$S_{t_1} = S_{t_0} + (t_1 - t_0) \cdot \frac{\delta S}{\delta t} = S_{t_0} + n \cdot \frac{\delta S}{\delta t} \qquad , \tag{1}$$

where $S_{t_0}$ is the regional glacier area in the (regionally averaged) survey year of the RGI, $\delta S/\delta t$ the annual area-change rate, and $n$ the number of years between $t_0$ and $t_1$.

**2.2 Regional glacier mass changes**

We use the regional and global glacier mass changes (based on spatial interpolation) from 1961/62 to 2015/16 from Zemp et al. (2019) as a reference dataset, including corrected values for Iceland (cf. Zemp et al., 2020). For each region, this data set combines the temporal variability from the glaciological sample, obtained using a spatiotemporal variance decomposition model, with the glacier-specific change rates of the geodetic sample. These calibrated annual time series in the metre water equivalent unit (1 m w.e. = 1,000 kg m$^{-2}$) were extrapolated from the observational to the full glacier sample within the region and multiplied by regional surface areas, resulting in regional mass changes in the unit Gt (1Gt = 10$^{12}$ kg). Full details are found in Zemp et al. (2019).

**2.3 Glaciological observations**

The glaciological method provides glacier-wide mass changes by using point measurements from seasonal or annual *in situ* campaigns, extrapolated to the overall glacier surface (cf. Cogley et al., 2011). We used the latest release of the Fluctuations of Glaciers (FoG) database as available from the WGMS (2019). This dataset is basically consistent with the glaciological data from 1961/62 to 2015/16 as used by Zemp et al. (2019) but includes updated mass balances for 2016/17 and 2017/18, as well as some corrections and addenda for earlier years. In total, this dataset contains 6,986 annual mass balances from 459 glaciers. For 2016/17 and 2017/18, it contains annual balances from 154 and 103 glaciers, respectively. The WGMS provides glaciological balances for hydrological years, which begin near the start of the accumulation season and end near the end of the ablation season (cf. Cogley et al., 2011). As a consequence, the results refer to different time periods when comparing regions from the Northern to the Southern Hemisphere, or to the Low Latitudes.

**2.4 *Ad hoc* estimation of regional mass changes and sea-level equivalents**

A change in climatic factors is reflected in a corresponding change of the (regional climatic) Equilibrium Line Altitude (ELA, cf. Cogley et al., 2011), which shifts the vertical mass-balance profile (Fig. S1a–c). The glaciers of a region can react with a large range of specific mass balances to such a change (Fig. S1d; Kuhn et al., 1985). At the same time, these glaciers are expected to feature common mass-balance anomalies (Fig. S1d; Vincent et al., 2017), i.e., positive or negative deviations for a decrease or increase of the ELA, respectively. Building on these basic assumptions, we calculated the annual *ad hoc* estimate for regional mass changes (Fig. S1e) and corresponding sea-level equivalents for a given *ad hoc* year of observations $Y$ (e.g. 2017/18) in the following five steps.

(*i*) For each glacier $g$ with observations in a given year $Y$, we calculated the glaciological mass-balance anomaly $\beta$, similar to the "centred mass balance" by Vincent et al. (2017), as the anomaly of the glaciological balance of the *ad hoc* year $B_{\text{glac},Y,g}$ with respect to the arithmetic mean balance over the calibration period from 2006/07 to 2015/16 $\bar{B}_{\text{glac},2007-2016,g}$:

$$\beta_{Y,g} = B_{\text{glac},Y,g} - \bar{B}_{\text{glac},2007-2016,g} \quad . \tag{2}$$

Note that $\sum_{i=2007}^{2016} \beta_{i,g} = 0$ by definition. Here, the calibration period was set to the last decade of available reference data as these years best reflect the current mass-change conditions and provide largest glaciological sample size.

(*ii*) For each RGI region $r$, the mean glaciological mass-balance anomaly $\bar{\beta}$ was calculated as the arithmetic average anomaly of the number of glaciers $G$ with available data in $Y$:

$$\bar{\beta}_{Y,r} = \frac{\sum_{g=1}^{G} \beta_{Y,g}}{G} \quad . \tag{3}$$

(*iii*) For each region $r$, the *ad hoc* estimate of the annual specific mass change $B_{\text{adhoc}}$ (in m w.e.) was calculated by adding the regional anomaly $\bar{\beta}$ of the *ad hoc* year (in m w.e.) to the mean specific mass change $\bar{B}_{ref}$ of the corresponding region from the reference data over the calibration period from 2006/07 to 2015/16 (in m w.e. yr$^{-1}$):

$$B_{\text{adhoc}.Y,r} = m \cdot \bar{\beta}_{Y,r} + \bar{B}_{\text{ref},2007-2016,r} \quad . \tag{4}$$

Basically, this corresponds to a linear regression model with slope $m$ and y-intercept $\bar{B}_{ref}$ (Fig. S2). We set $m = 1$ in order to make the *ad hoc* estimation applicable to reference data providing mass-change rates but without annual resolution (cf. Fig. S3) and compare these results with the ones from a regression model with variable slopes (Table S2). Apart from that, we

note that for a stable glacier sample (i.e., observations available from the same glaciers from year to year), the present approach corresponds to a simple bias correction of the glaciological sample with respect to the reference data. The corresponding regional bias $\varepsilon$ of the glaciological sample can be calculated as:

$$\varepsilon_{\text{glac},2007-2016,r} = \bar{B}_{\text{glac},2007-2016,r} - \bar{B}_{\text{ref},2007-2016,r} \quad . \tag{5}$$


(*iv*) For each region $r$, we calculated the *ad hoc* estimate of the regional mass change $\Delta M$ by multiplying the regional specific mass change by the regional glacier area for that particular year $S_{Y,r}$, considering the cumulative area changes since the survey year of the RGI (cf. Eq. 1):

$$\Delta M_{\text{adhoc},Y,r} = B_{\text{adhoc},Y,r} \cdot S_{Y,r} \quad . \tag{6}$$

(*v*) Finally, we calculated the *ad hoc* estimate of the corresponding worldwide sea–level equivalent *SLE* as:

$$SLE_{\text{adhoc},Y} = (-1 \cdot \sum_{r=1}^{R} \Delta M_{\text{adhoc},Y,r})/S_{\text{ocean}} \qquad , \qquad (7)$$

where $S_{\text{ocean}}$ is the area of the ocean with 362.5 x $10^6$ km$^2$ (Cogley, 2012).

For regions with no glaciological observations in the *ad hoc* year, we used available data from neighbouring regions. In line with Zemp et al. (2019), we selected WGMS *reference* glaciers with long-term data series from neighbouring regions that feature a similar mass-balance variability based on qualitative and quantitative criteria, such as a good correlation between mass-balance series available from earlier years (see Table S1).

## 2.5 Uncertainty estimates

The regional mass changes from Zemp et al. (2019) come with error bars considering uncertainties from four independent sources: the temporal variability in the glaciological sample, the long-term geodetic mass changes, the extrapolation to unmeasured glaciers, and the regional glacier area. We combined these overall error bars from Zemp et al. (2019) with an additional uncertainty related to the estimation of the mass-balance anomaly. For the latter, we estimated the uncertainty as 1.96 times the (sample) standard deviation of the mean glaciological mass-balance anomaly for each region over the calibration period from 2006/07 to 2015/16 (cf. Eq. 3), which corresponds to a 95% confidence interval. In cases with only one glacier in the glaciological sample (resulting in a standard deviation of zero), we set the uncertainty to 100% of the anomaly. The two errors related to the reference dataset and to the mass-balance anomaly were combined according to the law of random error propagation. For global sums, the overall error was calculated by cumulating the regional errors according to the law of random error propagation for independent terms.

## 3 Results and discussion

### 3.1 *Ad hoc* estimates for 2016/17 and 2017/18

For 2016/17, the glaciological observations from 154 glaciers (from 15 of 19 regions) give a global average specific mass change of –0.5 ± 0.4 m w.e. (Table 1). The above presented *ad hoc* estimation suggests a global mass change of –316 ± 240 Gt corresponding to 0.9 ± 0.6 mm SLE. Glaciers suffered most in Central Europe, Alaska, and in the Low Latitudes with regional specific mass changes being more negative than –1 m w.e. (Table 1, Fig. 1). The largest annual contributions to global sea-level originated from Alaska (–115 Gt), the Antarctic (–64 Gt), and High Mountain Asia (–41 Gt). For New Zealand, the Southern Andes, and Arctic Canada North, investigators reported positive mass balances resulting – after bias correction – in a net mass gain of 4 Gt (essentially from the latter region).

For 2017/18, data reported thus far from 103 glaciers (from 14 out of 19 regions) comprises about two third of the currently observed glaciers (Table S1). This is related to the one-year retention period that is granted to allow investigators time to properly analyse, document, and publish their data before submission to the WGMS. Based on this preliminary data, the global average specific mass change was –0.8 ± 0.3 m w.e. (Table 1). The *ad hoc* estimation results in a global mass change

of –502 ± 197 Gt, corresponding to 1.4 ± 0.5 mm SLE. Reported mass balances were negative in all regions. *Ad hoc* estimates indicate that specific mass balances were more negative than –1 m w.e. in six regions, with New Zealand and Alaska even exceeding –2 m w.e. (Table 1, Fig. 1). With respect to sea-level rise, Alaska (–190 Gt) and the Canadian Arctic (North and South combined: –129 Gt) were the largest contributors in this year.

The *ad hoc* estimates for 2016/17 and 2017/18 indicate both an annual global glacier contribution of about one millimetre SLE – a benchmark that was only exceeded three times during the period from 1961/62 to 2015/16, when compared to the reference data (Fig. 2). In fact, 2017/18 featured the largest mass loss of the entire data series. The latest glaciological observations, hence, provide evidence of continued increasing global glacier mass losses since the 1980s.

## 3.2 Comparison to global reference datasets

We calculated annual *ad hoc* estimates for all years back to 1961/62 using the period from 2006/07 to 2015/16 for anomaly calculation and bias correction (Fig. 2a). These can be compared to the global reference data over the 45 years before 2006/07. We note that this is not a validation against independent data but an approach to test the ability of the glaciological sample of a given year to estimate the global glacier contribution to sea-level rise. Overall, the *ad hoc* estimates are in good agreement with the reference data but feature a slightly larger variability. The latter can be partially explained by the smaller

sample size available for the *ad hoc* estimates. The best agreement is found over the more recent period back to 1990, followed by a strong variability and corresponding over and underestimations during the 1980s, and relative good agreement again in the 1970s and 1960s. The strongest deviations occur in 1963/64, 1964/65, and between 1977/78 and 1987/88 and seem to be correlated with years in which many regions had positive glaciological balances. At the same time, the variance decomposition model as used by Zemp et al. (2019) tends to reduce the variance for statistically small samples since it only

extracts the common year-to-year variability found in all glaciological time series of a region. The variability (at each glacier) that is not found at other locations is assigned to the residual (i.e., the unexplained variance). Therefore, our *ad hoc* estimate is generally well suited to assess the global value of the more representative reference data. However, in years with small data samples and strong anomalies it remains arguable which of the two approaches better represents the correct global glacier mass changes. The uncertainty range of the *ad hoc* estimates is larger than that of the reference data in the most

recent validation period, since it combines the error bars of the reference data with those from the bias estimates. For the earlier validation periods, the uncertainty range of the reference data gets larger whereas the one from the *ad hoc* estimates is still based on reference data from the calibration period. This is arguably an artefact from the optimization of our approach to the estimation of mass changes for the most recent years 2016/17 and 2017/18.

The use of Zemp et al. (2019) as reference dataset has the advantage of analysing the performance of the *ad hoc* estimation at annual time resolution back to the 1960s, and allows for adjusting the reference period. We show that the *ad hoc* estimate can be sensitive to the choice of the reference period (i.e., 2006/07–2016/17 in Fig. 1 and 2003/04–2008/09 in Fig. S3a–e), especially when it results in major changes in the glaciological sample such as the use of glaciers from neighbouring regions. As an example, the annual *ad hoc* estimates for Arctic Canada North change from +8 Gt and –93 Gt (Fig. 1c) to +6 Gt and –69 Gt (Fig. S3a) for 2016/17 and 2017/18, respectively. However, our approach can be applied to other reference datasets that provide regional mass changes for a multi-year period but without annual resolution. In Fig. S3, we demonstrate this with *ad hoc* estimations for selected regions (with large glacierization but limited data coverage in Zemp et al. (2019) and different reference datasets (Bolch et al., 2013; Gardner et al., 2013; Wouters et al., 2019). The relative difference between the anomalies (derived from the glaciological sample) of the two *ad hoc* years are consistent but the absolute values vary between the different reference datasets (due to different mass change rates over the calibration periods). This implies that our approach allows for a regional selection of reference datasets and, hence, can be used in future consensus estimates of global glacier mass changes.

### 3.3 Lessons learned for the glaciological observation network

In the field of glacier monitoring, one outstanding question is how representative the local glaciological observations for regional to global mass changes are (Fountain et al., 2009; Kaser et al., 2006). With the availability of a global reference dataset Zemp et al. (2019), the present approach allows us assessing the bias in the glaciological observations for all regions (cf. Eq. 5, basically the difference between $B_{glac}$ and $B_{adhoc}$ in Table 1). At a regional level, the annual bias ranges between –0.6 and +0.5 m w.e. This confirms that the glaciological observations are well suited to represent the temporal variability but not necessarily the absolute value of regional glacier mass changes. At global level, the bias averaged out for the area-weighted mean. However, this is rather fortuitous and can change with the use of a different reference dataset (e.g., Gardner et al., 2013; Wouters et al., 2019).

Another question is whether a glaciological observation network reduced to the long-term observation series is good enough to estimate the temporal variability of global glacier mass changes. We thus performed another *ad hoc* estimate with a glaciological sample reduced to the current 41 WGMS *reference* glaciers (Fig. 2b), coming with more than 30 years of ongoing mass-balance measurements (WGMS, 2017). The WGMS *reference* glaciers provide a more stable sample over time but come at the price of a much reduced sample size. As such, the sample is reduced from 154 to 38 glaciers with observations in 2016/17 and, hence, neighbouring glaciers are needed in 10 (instead of 4) out of 19 regions (Table S1). The *ad hoc* estimates for the WGMS *reference* glacier sample from 1975/76 (i.e., the first year with *reference* glacier data from the southern hemisphere) to 2016/17 show a much-increased variability, strong offsets over certain periods, and an increased uncertainty by about 40% (Fig. 2, Fig. S4). This low performance suggests that – for the present approach – the WGMS

*reference* glacier sample alone is too small and represents too few regions for an *ad hoc* estimation of global glacier contributions to sea-level rise. It is worthwhile to note that regions with large areas of glacierization (e.g. Arctic Canada South, Russian Arctic, South Asia East & West, peripheral Greenland, and peripheral Antarctica) lack long-term mass-balance series.

**4 Conclusions and outlook**

Direct glaciological observations, as currently conducted for about 150 glaciers worldwide, are able to satisfactorily capture the temporal mass-balance variability but are often not representative of the total mass change of a region. We presented a new approach to provide *ad hoc* estimates of regional glacier mass changes for the most recent years based on the anomaly of glaciological mass-balance observations and a bias correction to a reference dataset over a common calibration period
from 2006/07 to 2015/16. The *ad hoc* estimates for 2016/17 and 2017/18 indicate that global glacier mass loss has further increased (with respect to the previous decade) and resulted in annual global glacier contributions to sea-level rise exceeding 1 mm SLE, which corresponds to more than a quarter of the currently observed sea-level rise (cf. IPCC, 2019). Our new approach allows for the timely reporting of global glacier mass changes and can be applied to a new consensus estimate as reference data, once available. To increase the accuracy of the global *ad hoc* estimates, we need to extend the glaciological
sample into so far underrepresented and strongly glacierized regions such as High Mountain Asia, the Southern Andes, Russian Arctic, Greenland or Antarctica. At the same time, we need to tap the full potential of spaceborne surveys to further improve the spatio-temporal coverage and resolution of the reference datasets.

**Code availability**

The analytical scripts are available from the lead author on request.

**Data availability**

The full sample of glaciological observations for individual glaciers is publicly available from the WGMS (https://doi.org/10.5904/wgms-fog-2019-12). The regional and global reference datasets from Zemp et al. (2019) are available from the Zenodo repository (https://doi.org/10.5281/zenodo.3557199).

**Supplementary material related to this article is available online at: [URL]**

**Author contribution**

MZ, FP, and MH developed the basic concept of the study consulting with ET and NE for statistical backup; MZ, IG, and SN compiled and quality-checked the glaciological data with the support of the WGMS collaboration network; MZ performed all computations, designed the figures and wrote the manuscript. All authors commented on the manuscript.

**Competing interests**

The authors declare no competing interests.

**Acknowledgements**

We are indebted to the national correspondents and principal investigators of the WGMS for sharing their observations with the community. We thank R. Hock for constructive input on mass-balance terminology and units, and B. Armstrong for polishing the language. This study was enabled by financial support from the Federal Office of Meteorology and Climatology MeteoSwiss within the framework of the Global Climate Observing System (GCOS) Switzerland, the Copernicus Climate Change Service (C3S) implemented by the European Centre for Medium-range Weather Forecasts (ECMWF) on behalf of the European Commission, the ESA project Glaciers_cci (4000109873/14/I-NB), and Irstea Grenoble as part of LabEx OSUG@2020.

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

**Table 1** *Ad hoc* estimates of glacier mass changes in 2016/17 and 2017/18. For both years, the table shows glacier areas (S) based on RGI 6.0 (2017) and corrected for annual area change rates from Zemp et al. (2019), specific mass changes calculated as arithmetic mean of the glaciological sample ($B_{glac}$) and based on the *ad hoc* estimation ($B_{adhoc}$, Eq. 4), as well as anomaly-corrected mass change ($\Delta M$, Eq. 6) for all regions and global totals. Global specific mass changes and related biases are calculated as area-weighted regional means. Uncertainties correspond to 95% confidence intervals. The annual global mass changes in 2016/17 and 2017/18 correspond to $0.9 \pm 0.6$ and $1.4 \pm 0.5$ mm SLE, respectively. Glaciological input data are from WGMS (2019).

| Region | Area (km²) | | Specific mass change (m w.e.) | | | | Mass change (Gt) | |
|---|---|---|---|---|---|---|---|---|
| | $S$ 2017 | $S$ 2018 | $B_{glac}$ 2017 | $B_{adhoc}$ 2017 | $B_{glac}$ 2018 | $B_{adhoc}$ 2018 | $\Delta M$ 2017 | $\Delta M$ 2018 |
| 01 Alaska | 83,395 | 82,978 | −1.18 | −1.37±0.63 | −1.85 | −2.29±0.56 | −115±53 | −190±47 |
| 02 Western Canada & USA | 13,661 | 13,583 | −0.54 | −0.68±1.05 | −0.65 | −0.85±0.74 | −9±14 | −12±10 |
| 03 Arctic Canada North | 103,860 | 103,787 | 0.03 | 0.07±0.99 | −0.82 | −0.90±0.87 | 8±103 | −93±91 |
| 04 Arctic Canada South | 40,332 | 40,299 | −0.22 | −0.22±1.18 | −0.82 | −0.90±0.78 | −9±48 | −36±31 |
| 05 Greenland | 77,946 | 77,210 | −0.41 | −0.27±0.65 | −0.33 | −0.44±0.60 | −21±50 | −34±46 |
| 06 Iceland | 10,383 | 10,343 | −0.29 | −0.11±0.76 | −0.05 | 0.14±0.99 | −1±8 | 1±10 |
| 07 Svalbard & Jan Mayen | 32,546 | 32,458 | −0.59 | −0.57±0.64 | −0.62 | −0.69±0.46 | −18±21 | −23±15 |
| 08 Scandinavia | 2,830 | 2,822 | −0.03 | −0.09±1.35 | −1.44 | −1.48±0.86 | 0±4 | −4±2 |
| 09 Russian Arctic | 51,138 | 51,097 | −0.72 | −0.69±0.45 | −0.82 | −0.80±0.45 | −35±23 | −41±23 |
| 10 North Asia | 2,348 | 2,337 | −0.90 | −0.67±0.76 | −0.39 | −0.17±0.64 | −2±2 | 0±2 |
| 11 Central Europe | 1,820 | 1,800 | −1.64 | −1.60±0.81 | −1.44 | −1.43±1.06 | −3±2 | −3±2 |
| 12 Caucasus & Middle East | 1,196 | 1,189 | −0.84 | −0.89±0.90 | −0.22 | −0.28±1.67 | −1±1 | 0±2 |
| 13 Central Asia | 48,061 | 47,972 | −0.77 | −0.39±0.76 | −0.51 | −0.11±0.59 | −19±37 | −5±29 |
| 14 South Asia West | 31,876 | 31,755 | −0.90 | −0.34±0.69 | −0.39 | 0.17±0.55 | −11±22 | 5±18 |
| 15 South Asia East | 13,765 | 13,695 | −1.07 | −0.77±1.02 | −1.37 | −1.10±0.44 | −11±14 | −15±6 |
| 16 Low Latitudes | 1,867 | 1,840 | −1.06 | −1.13±1.53 | −0.37 | −0.29±1.78 | −2±3 | −1±3 |
| 17 Southern Andes | 28,528 | 28,476 | 0.37 | −0.13±4.43 | −0.64 | −1.11±2.11 | −4±126 | −32±60 |
| 18 New Zealand | 849 | 841 | 0.41 | 0.13±1.78 | −2.34 | −2.62±1.77 | 0±2 | −2±2 |
| 19 Antarctic & Subantarctic | 122,822 | 122,464 | −0.28 | −0.52±1.16 | −0.13 | −0.16±1.14 | −64±143 | −20±140 |
| Global total, excl. 05 & 19 | 468,455 | 467,272 | −0.53 | −0.49±0.40 | −0.91 | −0.96±0.28 | −231±186 | −449±131 |
| Global total | 669,223 | 666,946 | −0.47 | −0.47±0.36 | −0.70 | −0.75±0.29 | −316±240 | −502±197 |

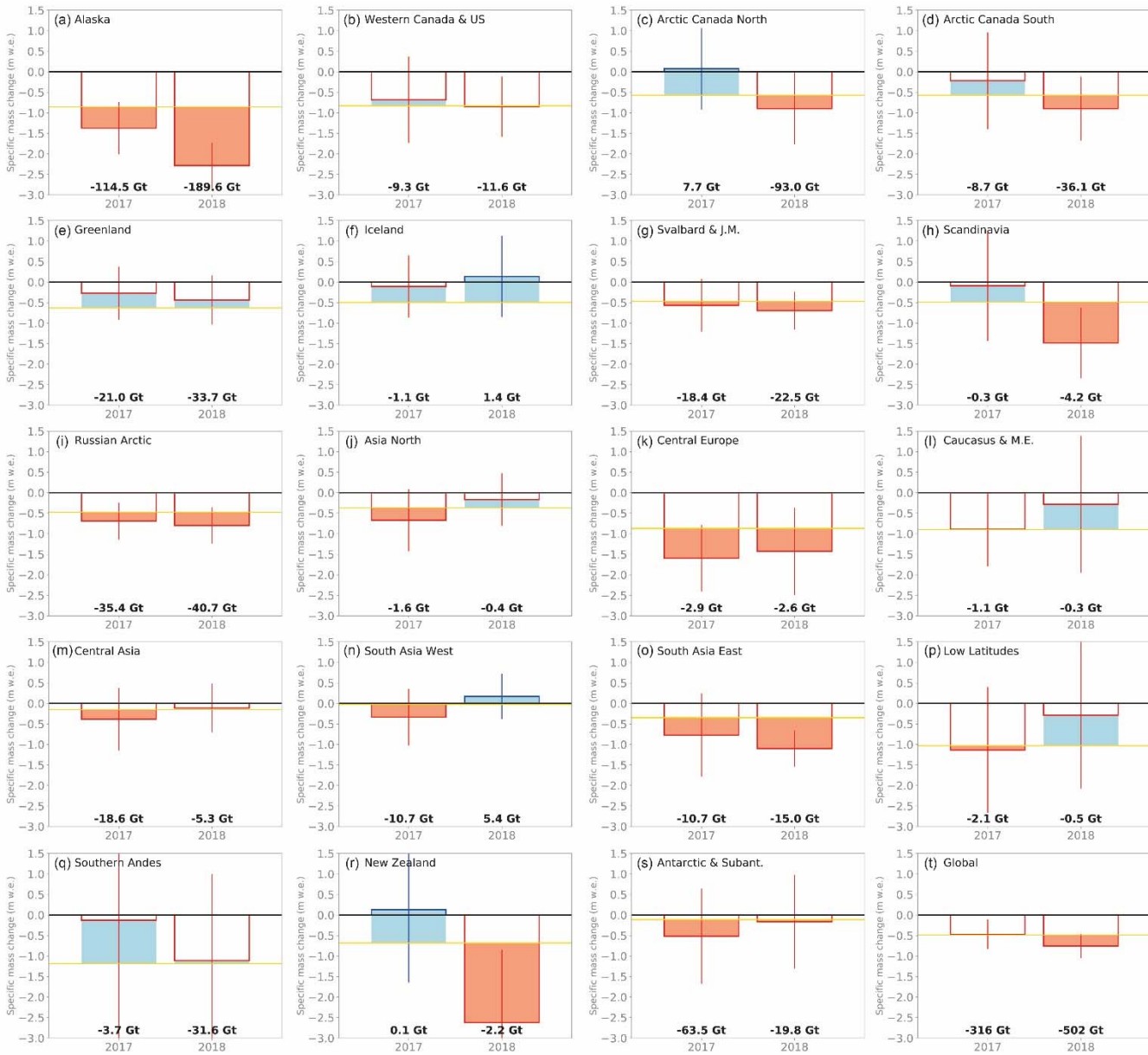

**Figure 1** *Ad hoc* estimates of regional mass changes in 2016/17 and 2017/18. The regional (**a–s**) and global (**t**) bar plots show the annual specific mass changes (in m w.e.) with related error bars (indicating 95% confidence intervals), with positive and negative values in blue and red, respectively. The golden line indicates the annual mass-change rate of the reference data (in m w.e. yr$^{-1}$; Zemp et al., 2019) over the calibration period (2006/07–2015/16). Positive and negative annual mass-change anomalies (with respect to reference data and calibration period) are indicated in pale blue and pale red, respectively. The black values (bottom) indicate annual mass changes in Gt. Plots are ordered from top left to bottom right according to the region numbers in RGI 6.0 (see Table 1).

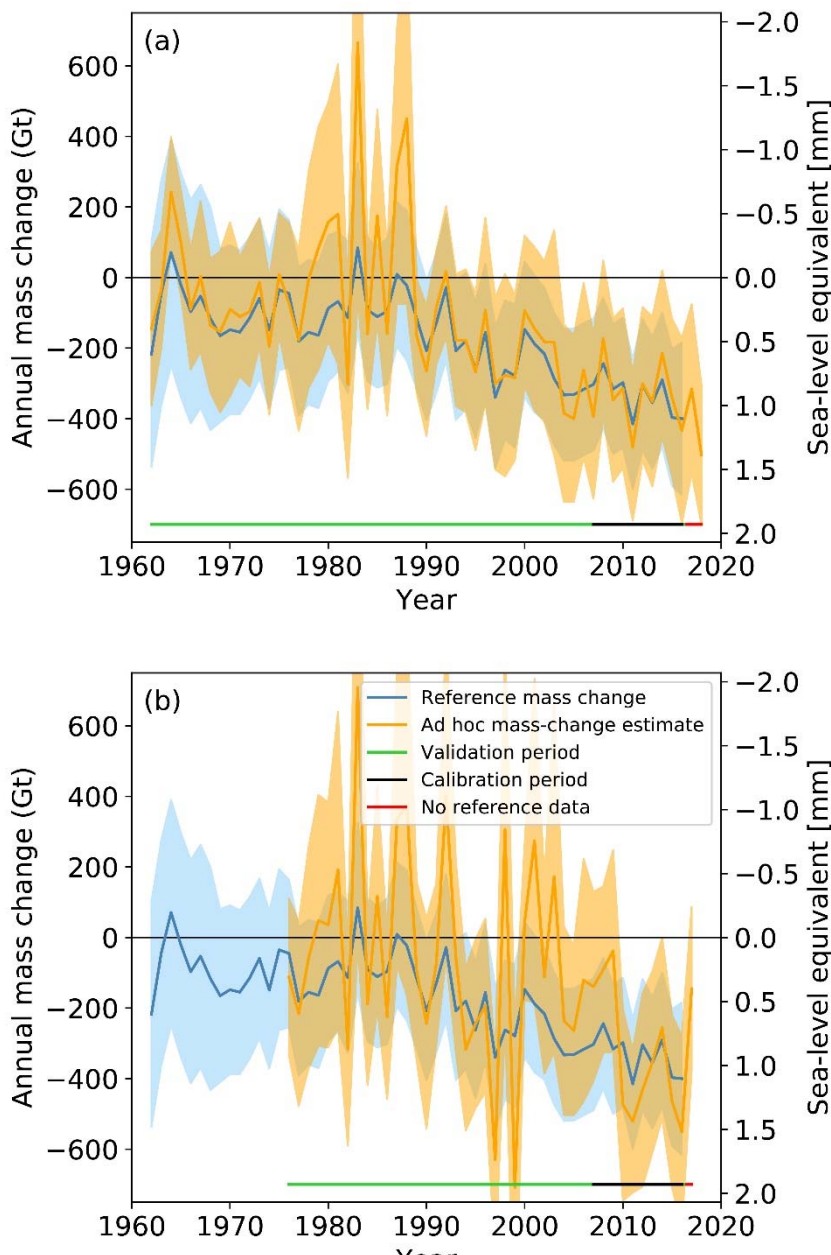

**Figure 2** Global glacier contributions to sea-level rise from 1961/62 to 2017/18. (**a**) Full glaciological sample: annual mass-changes (left y-axis) and global sea-level equivalents (right y-axis) are shown with related error bars (indicated by shadings) corresponding to 95% confidence intervals. The *ad hoc* estimates (orange), based on the full glaciological samples of corresponding years, are shown with annual values of the reference dataset (blue) by Zemp et al. (2019), which was used for calibration. The bottom line indicates the time periods used for calibration (black), validation (green), and without reference data (red). (**b**) *Reference* glacier sample: same plot but for *ad hoc* estimates (orange) solely based on glaciological data from the 41 WGMS *reference* glaciers (with more than 30 years of ongoing measurements). Due to limited data coverage, no *ad hoc* estimates were possible for 2017/18 and before 1975/76.

