# Peer review of "Brief communication: *Ad hoc* estimation of glacier contributions to sea-level rise from latest glaciological observations"

_The Cryosphere, 2019_

## Referee Comment (RC1) · Anonymous Referee #1 · 7 Oct 2019

**Review of 'Brief communication: Ad hoc estimation of glacier contributions to sea-level rise from latest glaciological observations' by Zemp et al. (2019, The Cryosphere Discussions, doi: 10.5194/tc-2019-180)**

In this manuscript, Zemp and co-authors introduce a new method to derive 'ad hoc' estimates of regional glacier loss based on a sample of glaciological observations. They use the method to provide estimates for the regional glacier mass changes for 2016/17 and 2017/18 and sum these regional values to obtain an estimate of the total sea-level contribution from glaciers for these years.

The method introduced and described by Zemp and colleagues is simple and efficient, and is well described in this brief communication. It will be of great use when determining the sea level contribution from glaciers for recent years, which will be particularly useful when providing up-to-date numbers e.g. for the upcoming IPCC sixth assessment report. The contribution is thus timely, although it would be interesting if the data included for the year 2017/18 could somewhat be extended (more on this below). I do have some questions and suggestions that could improve the clarity of the manuscript, but these are generally relatively minor. My questions/suggestions have therefore been arranged per section, and not by 'major', 'minor' and 'technical' comments. The list may seem relatively long at first, but the vast majority of comments should be easy to address.

**Abstract**

- p.1, l.12: 'the glaciological method': clear for people who are in the field of glacier mass balance, but not straightforward for other glaciologists. Would maybe be good if you can describe what the 'glaciological method' is in a few words
- p.1, l.15: Another clarification here, what does 'ad hoc estimate' mean: again not sure that this would be directly understandable. Also given the fact that it appears in the title, would be good to explain shortly: this 'ad hoc' estimate refers to the fact that this is a kind of 'on-the-go' / 'specific' / 'best given the available data' / …. estimate (what you think describes it best)?

**Introduction**

- p.1, l.20: 'substantially contribute to regional runoff': could also add references to two important new regional studies focusing river runoff and the role of glaciers: Biemans et al. (2019) and Pritchard (2019)
- p.1, l.28-29: 'In view of the IPCC Special Report on the Ocean and Cryosphere in a changing climate (2019)' → has been released by now: add a reference to SPM (IPCC, 2019) or specific chapter
- p.2, l.2: 'for the comparison with estimates based on other methods (e.g. spaceborne gravimetry or altimetry)': could you be a more specific here? Which studies are you referring to?
- p.2, l.4: 'In view of the global stocktake': what is this?
- p.2, l.5-8: 'the approaches underlying these results are unsuitable for providing annual updates on the basis of new glaciological observations acquired each year due to generic lack of updates from multi-annual geodetic surveys (from DEM differencing)': OK, this is true. But is likely to change in the coming years, with DEMs becoming more reliable (more precise and with less errors and artefacts) and available more regularly (likely at sub-annual intervals in the near future). Would be good if could comment on this and put this a bit in perspective: probably the conclusion, where you stress the importance of continued field observations, would be good to do this
- p.2, l.8: 'a computational framework': sounds like a (complex) numerical model you are using. Would suggest omitting the 'computational'

**Data and methods**

- p.2, l.26: 'regional area in the survey year of the RGI': how is the survey year for a given region defined? Reason why I ask is because some of the RGI regions do not have a single survey year, but consist of outlines derived over various years, right?
- p.2, l.26: 'delta-S/delta-t the annual area-change rate': could you provide a hint about how this is determined? I understand that not all elements from Zemp et al. (2019) can be repeated for

the sake of brevity, but here would be nice to be able to have an insight without having to look into the other paper.

- p.3, l.22: 'anomaly from the mean balance over the calibration period from 2006/07 to 2015/16': not sure I entirely understand. Do you refer to the anomaly 'compared to' or 'with respect to' 'the mean balance….'? Consider reformulating to make this clearer.
- p.3, l.22: the reference period is chosen as 2006/07 to 2015/16: could you put this in perspective? i.e. why is this 10-year reference period chosen and why not for instance a longer time period (e.g. 15 to 20 year period) or another 10-year period? Is this mainly related to the fact that glaciers have changed a lot, or is this maybe related to the sample size when going further back in time?
- p.3, l.22: a follow-up question: what influence does the choice of the reference time period have? Would for instance be interesting to see how the sea level contributions for 2016/17 and 2017/18 are affected by the choice of the reference period (i.e. get insight in the sensitivity of your results to the reference time period choice).
- p.3, l.25: equation 2: are $B_{glac,Y,g}$ and $B_{glac,g,i}$ defined?
- p.3, l.26: 'results show that equation=0': formulation sounds a bit strange to me, as I suppose this is a direct consequence of how it is defined (and not really a result). Consider reformulating this: 'Over the calibration period equation=0' (i.e. omit the 'results show that')
- p.3, l.31: 'G': G corresponds to number of glaciers?
- p.4, l.1: equation 3. can you explain why it is not weighted by glacier area? Would intuitively expect this, but probably related to misunderstanding from my side. May be good to shortly explain in text also.
- p.4, l.14: equation 5: where is this 'regional bias of the glaciological sample' used later on?
- p.4, l.16: 'regional glacier area S': may be good to specify that this is the value for that particular year. This becomes somewhat clear later in the sentence, but nevertheless good to stress this to avoid any possible confusion.
- p.4, l.26-27: you explain that reference glacier from neighbouring region is used when no glaciological observations are available for a given region. Little data, so probably not many good options, but this is nevertheless a rough approximation, especially given the distance between the region. Three questions here:
    o How do you define which one is the neighbouring region when there are several options?
    o How large is the effect of choosing glaciers from another region: could you quantify this? e.g. with a kind of 'leave-one-out cross validation'?
    o Is there no better criterion than proximity of the regions to fill the gap? i.e. would it for instance make sense to determine which regions correlate best for a given reference period in the past (which may not always be the neighbouring I guess) and use this information to fill up the gaps?
- p.5, l.2: 'For each region, we calculated the…': here, or in Zemp et al. (2019)?
- p.5, l.2-3: '1.96 time the (sample) standard…': why 1.96?

**Results and discussion**

- Global average specific mass change of -0.5 and -1.0 m w.e. yr$^{-1}$ for 2016/17 and 2017/18 respectively. Uncertainty on these values? Same relative uncertainty as for the global mass change in Gt yr$^{-1}$ and the sea-level contribution?
- p.5, l.18-25:
    o Only little data available for 2017/18. Is well explained why that is, but a pity that is a somewhat incomplete dataset. Would be nice if this could be updated when finalising this manuscript, with the latest data included. Numbers will be used (potentially copied without context?) by other scientists and would therefore be good if these are (close to) the final numbers (i.e. with more glaciers).
    o It is interesting to see how the method works with different sample sizes, but from the setup used so far it is difficult to estimate what the influence is on the regional mass balances, the global sea level contribution and its uncertainty. Would be very useful if some experiments would be performed to quantify this: e.g. taking only 20-40-60-80- ..glaciers and to see where is it crucial to have data: i.e. which regions influence the final numbers the most and where it is problematic (large increase in uncertainty) when data lacks?

- p.5., l.11 & l.21: uncertainty on the global mass change is substantially lower for 2017/18 (138 Gt yr$^{-1}$) compared to 2016/17 (249 Gt yr$^{-1}$). Why is this? Would intuitively expect the opposite, given the limited number of glaciers considered in 2017/18 (70) vs. 2016/17 (150). This comment is related to previous suggestions to quantify the effect of glacier sample size and the effect of the location/region on the uncertainty.
- p.5, l.21: mass loss of 512 Gt yr$^{-1}$: would reformulate to mass change of -512 Gt yr$^{-1}$ to be consistent with formulation for 2016/17 and avoid any confusion. My first reaction when reading this was: 'oh, a positive mass balance?' and then I re-read it and saw that here you refer to a loss vs. a mass change.
- p.5, l.25: '…contributor**s**.'
- p.6, l.9: 'again relative good agreements again in the…' (omit one of the two again's)
- p.6, l.14-15: 'in years with small data samples and strong anomalies it remains arguable which of the two approaches better represents…': would it be feasible to quantify this? Would be interesting addition.
- p.6, l.28: at the end of this paragraph: would expect that you explain what the consequence of this statement is: e.g. "This does imply that…"
- p.7, l.7: 'regional biases range between -0.6 and +0.5 m w.e. yr$^{-1}$': this is quite large, in the order of the signal almost, no?
- p.7, l.5-16: ok, interesting! But what if you work with a method more sophisticated than taking values from neighbouring regions? (see earlier comment on this)

**Conclusion**

- Explains nicely why this work is important and why we need such updates. When you present contributions to sea level (p.7, l.23-24), could you compare these numbers to the contribution from the ice sheets and thermal expansion? (for these 2016/17 and 2017/18 specifically, and if not available, with the numbers from the last years/decade) Would be good to stress, once again, the important sea-level contribution from glaciers, which ice sheet modellers quite often tend to forget/underestimate..
- p.7, l.25-28: importance of glaciological sample is stressed. Fully agree and think that this study nicely supports this statement. It would however also be justified to put this a bit in perspective and explain that in the future we will be able to rely more and more on remote sensing observations. These observations will become available at higher resolution, with smaller uncertainties and biases, and most importantly: at a high temporal resolution (with sub-annual update). As such, they could therefore be used in addition to glaciological measurements (still a long way to go for replacing them…), making us less dependent on direct field measurements.

**References**

Biemans, H., Siderius, C., Lutz, A. F., Ahmad, B., Hassan, T., von Bloh, W., et al. (2019). Importance of snow and glacier meltwater for agriculture on the Indo-Gangetic Plain. *Nature Sustainability*, *2*. https://doi.org/10.1038/s41893-019-0305-3

IPCC. (2019). Summary for Policymakers. In H.-O. Pörtner, D. C. Roberts, V. Masson-Delmotte, P. Zhai, M. Tignor, E. Poloczanska, et al. (Eds.), *IPCC Special Report on the Ocean and Cryosphere in a Changing Climate*.

Pritchard, H. D. (2019). Asia's shrinking glaciers protect large populations from drought stress. *Nature*, *569*, 649–654. https://doi.org/10.1038/s41586-019-1240-1

Zemp, M., Huss, M., Thibert, E., Eckert, N., McNabb, R., Huber, J., et al. (2019). Global glacier mass balances and their contributions to sea-level rise from 1961 to 2016. *Nature*, *568*, 368–386. https://doi.org/10.1038/s41586-019-1071-0

---

## Referee Comment (RC2) · Anonymous Referee #2 · 30 Oct 2019

This is a nice, simple, and clear paper, but I am not convinced about the uncertainty budget (p5 line1-5). This needs to be fixed or argued better.

You are assuming that there is a very simple relationship between $\beta$avg and B. Basically a line with slope of 1 and an intercept that you estimate. I would like to see supplementary scatterplots of $\beta$avg vs B, with that line you estimate. One plot per region, with one point for every calibration year. You could also add the two ad-hoc estimates with error bars. This would be valuable because it instinctively gives you an idea if the slope=1 is a reasonable assumption, how large the scatter is, and give you an idea if you trust the ad-hoc estimates.

[Figure]

- paper structure, abstract, and conclusion: good.

Minor comments:

The equations are so simple that they are almost unnecessary. I would rather have an illustrative visualization explaining the method (one of the scatter plots).

p1 line 16: Why not the estimates to the abstract.

p3 line 30: Here you essentially assume that the arithmetic average of all the $\beta$'s is representative of the glaciers throughout the region. However, I imagine that $\beta$ will correlate with e.g. glacier size, and that the sample is not representative in terms of size (and other characteristics). I don't expect this to introduce a major bias, but the possibility should probably be considered when estimating the errors. I propose that you explicitly mention the assumptions at this point.

equation 2 and 3 (and...): It is good to be clear, but many of these equations are perhaps a bit over-verbose. These two eqns. basically just define what a mean value is.

p4 line 10: Is it a stable glacier sample? What do you mean by stable?

p4 lines 15-20 and equation 6: I assume that the 1/1000 is a unit conversion. Please remove this. The method is independent of units and we trust you to do the conversions correctly when you report the results.

p4 equation 7: Please remove the 10^6 unit-conversion multiplier.

p4 line 25: This is on the other hand important to specify.

p4 line 27: There is only one region with n.a. in table 1 — Please specify which region at this point.

p5 line 1-5: This method seems rather arbitrary. Is it really mathematically justified? E.g. this method has no consideration of how representative the sample is. Consider

also the case where the sample is complete (has the entire population). In this case you this error term should be zero (right?). What would your current 1.96x std method give?

Would it not be better to estimate the uncertainty from the residuals. e.g. using a jack-knife approach. You could withhold the data for one of the years in the 2007-2016 and see how well your method can predict the with held data. That would give you 10 residuals for each region. From these you can estimate both an RMS error and a mean bias. It would even work if your sample has only a single glacier.

p7 line 13-14: I would add "... using this method." to this sentence. Maybe another statistical treatment than your proposed method would have better performance.

equation 6: Does it have any practical effect to allow for changing area rather than just using a constant? It would make it even easier to get ad-hoc estimates, if you did not need new estimates of S.

p3 line 27: Is the sentence starting with "Over the calibration period ..." necessary? Isnt it clear it must be so, given the definition?

p2 line 4: "))"

p1 line 27: "AR 5"->"AR5"

p6 line 9: There is a limit to how far you can extrapolate the empirical calibration. The proposed scatterplots would help diagnosing the problem.

Figure S2: Please expand on the description in the caption and page 6 line24-26. It is not sufficient to understand what has been done.

---

## Author Comment (AC1) · 18 Dec 2019

We thank the two anonymous referees for their positive and thorough reviews. We revised our manuscript under consideration of all their feedback, which helped to improve the clarity of the brief communication.

In the following, we give a short response to the general comments by the reviewers:

Limited data availability for 2017/18: We re-computed all calculations with a newer release of the glaciological dataset as meanwhile available from the World Glacier Monitoring Service (DOI: 10.5904/wgms-fog-2019-12) with improved data coverage in

2017/18, and correspondingly updated all values, tables and figures. As a consequence, our ad hoc estimates are now based on similar sample sizes and provide mass change estimates for all glacier regions for both years 2016/17 and 2017/18. Overall this update resulted in minor changes of regional and global results well within error bars and support the applicability of the proposed method.

General assumptions and error budget: Several questions by the reviewers were related to our basic assumptions and to the uncertainty estimates for the glacier mass-balance anomaly. We decided to maintain our approach but introduced an additional paragraph in the method section (2.4) with a corresponding figure in the supplement (Fig. S1a–e) explaining our general assumptions. In addition, we clarified the section (2.5) about our uncertainty estimates and added an additional figure and table (Fig. S2a–s, Table S2) discussing our approach in more details.

Sensitivity to choice of reference period: In general, our approach is not supposed to be (very) sensitive to changes in the selection of the reference period: a possible change in the mass-balance trend of the reference data is (partly) compensated by a corresponding change in the mass-balance anomalies of the glaciological dataset. However, larger changes in the reference period will influence the results due to a change in the glaciological sample size (which is decreasing back in time) and, hence, can force the use of data from neighbouring regions. In the revised manuscript, we clarified the corresponding discussion section (3.2) and demonstrate the related effect by using a different reference periods in Fig. S3a–e as compared to Fig. 1.

A detailed point-by-point reply is provided in the supplement to this comment.

Please also note the supplement to this comment:
https://www.the-cryosphere-discuss.net/tc-2019-180/tc-2019-180-AC1-supplement.pdf
* * *
[Figure]

**Supplement:**

**Brief communication: *Ad hoc* estimation of glacier contributions to sea-level rise from latest glaciological observations**

Michael Zemp[1], Matthias Huss[2,3], Nicolas Eckert[4], Emmanuel Thibert[4], Frank Paul[1], Samuel U. Nussbaumer[1,3], Isabelle Gärtner-Roer[1]

[1]Department of Geography, University of Zurich, Zurich, 8057, Switzerland
[2]Laboratory of Hydraulics, Hydrology and Glaciology (VAW), ETH Zurich, Zurich, 8093, Switzerland
[3]Department of Geosciences, University of Fribourg, Fribourg, 1700, Switzerland
[4]Université Grenoble Alpes, Irstea, UR ETGR, Grenoble, 38402, France

*Correspondence to*: Michael Zemp (michael.zemp@geo.uzh.ch)

**Point-by-point reply to reviewer comments**: reviewer comments are in black and the authors' response in blue with citations from the revised manuscript in green.

REVIEWER #1

In this manuscript, Zemp and co-authors introduce a new method to derive 'ad hoc' estimates of regional glacier loss based on a sample of glaciological observations. They use the method to provide estimates for the regional glacier mass changes for 2016/17 and 2017/18 and sum these regional values to obtain an estimate of the total sea-level contribution from glaciers for these years.

The method introduced and described by Zemp and colleagues is simple and efficient, and is well described in this brief communication. It will be of great use when determining the sea level contribution from glaciers for recent years, which will be particularly useful when providing up-to-date numbers e.g. for the upcoming IPCC sixth assessment report.

Response: We thank the reviewer for the positive evaluation and the constructive feedback.

The contribution is thus timely, although it would be interesting if the data included for the year 2017/18 could somewhat be extended (more on this below).

Response: We re-ran all calculations with a newer release of the glaciological dataset (fog-2019-12), with better data coverage in 2017/18 (see also author's response to general comments), and correspondingly updated all values, tables and figures.

I do have some questions and suggestions that could improve the clarity of the manuscript, but these are generally relatively minor. My questions/suggestions have therefore been arranged per section, and not by 'major', 'minor' and 'technical' comments. The list may seem relatively long at first, but the vast majority of comments should be easy to address.

**Abstract**

§ p.1, l.12: 'the glaciological method': clear for people who are in the field of glacier mass balance, but not straightforward for other glaciologists. Would maybe be good if you can describe what the 'glaciological method' is in a few words

Response: We agree and added the following description to Section 2.3: "The glaciological method provides glacier-wide mass changes by using point measurements from seasonal or annual *in situ* campaigns, extrapolated to the overall glacier surface (cf., Cogley et al., 2011)."

§ p.1, l.15: Another clarification here, what does 'ad hoc estimate' mean: again not sure that this would be directly understandable. Also given the fact that it appears in the title, would be good to explain shortly: this 'ad hoc' estimate refers to the fact that this is a kind of 'on-the-go' / 'specific' / 'best given the available data' / …. estimate (what you think describes it best)?

Response: We agree and added a short definition to Section 1: "Here, we present a framework to infer *ad hoc* (i.e., timely but preliminary) estimates of global-scale glacier contributions to sea-level rise from annual updates of glaciological observations."

**Introduction**

§ p.1, l.20: 'substantially contribute to regional runoff': could also add references to two important new regional studies focusing river runoff and the role of glaciers: Biemans et al. (2019) and Pritchard (2019)

Response: We added these references to the introduction section.

§ p.1, l.28-29: 'In view of the IPCC Special Report on the Ocean and Cryosphere in a changing climate (2019)' à has been released by now: add a reference to SPM (IPCC, 2019) or specific chapter

Response: Done.

§ p.2, l.2: 'for the comparison with estimates based on other methods (e.g. spaceborne gravimetry or altimetry)': could you be a more specific here? Which studies are you referring to?

Response: Done.

§ p.2, l.4: 'In view of the global stocktake': what is this?

Response: The *global stocktake* is the process to assess the collective progress towards achieving the Paris Agreement. More details are found in the reference given: UNFCCC (2016).

§ p.2, l.5-8: 'the approaches underlying these results are unsuitable for providing annual updates on the basis of new glaciological observations acquired each year due to generic lack of updates from multi-annual geodetic surveys (from DEM differencing)': OK, this is true. But is likely to change in the coming years, with DEMs becoming more reliable (more precise and with less errors and artefacts) and available more regularly (likely at sub-annual intervals in the near future). Would be good if could comment on this and put this a bit in perspective: probably the conclusion, where you stress the importance of continued field observations, would be good to do this

Response: We agree that the spaceborne geodetic methods provide a great potential for delivering DEMs with improved resolution and quality for global glacier volume-change assessments. However, this will not automatically solve the general issues related to the timeliness of data availability (i.e., from DEM production to the publication of the results, to data submission and integration at individual glacier level into the international data repositories, and to open access for the community) and to density conversion (which become more severe for short survey periods, cf. Huss

2013, The Cryosphere). We extended the outlook statement in the conclusions as follows: At the same time, we need to tap the full potential of spaceborne surveys to further improve the spatio-temporal coverage and resolution of the reference dataset."

§ p.2, l.8: 'a computational framework': sounds like a (complex) numerical model you are using. Would suggest omitting the 'computational'

Response: We deleted "computational".

**Data and methods**

§ p.2, l.26: 'regional area in the survey year of the RGI': how is the survey year for a given region defined? Reason why I ask is because some of the RGI regions do not have a single survey year, but consist of outlines derived over various years, right?

Response: Yes, this is correct. As the outlines in RGI 6.0 come with a time stamp (i.e. BgnDate, EndDate) referring to the date (range) of the source from which the outlines were taken, we assigned the regional glacier area to the regional average survey year as listed in the RGI. We clarify the text as follows: "… $S_{t0}$ is the regional glacier area in the (regionally averaged) survey year of the RGI, …"

§ p.2, l.26: 'delta-S/delta-t the annual area-change rate': could you provide a hint about how this is determined? I understand that not all elements from Zemp et al. (2019) can be repeated for the sake of brevity, but here would be nice to be able to have an insight without having to look into the other paper.

Response: We agree and rewrote the text as follows: "We consider changes in glacier area over time by using annual change rates for all first-order regions from Zemp et al. (2019, and references therein), based on a data collection from the literature."

§ p.3, l.22: 'anomaly from the mean balance over the calibration period from 2006/07 to 2015/16': not sure I entirely understand. Do you refer to the anomaly 'compared to' or 'with respect to' 'the mean balance….'? Consider reformulating to make this clearer.

Response: Clarified (see response to comment on p.3, l.25 below).

§ p.3, l.22: the reference period is chosen as 2006/07 to 2015/16: could you put this in perspective? i.e. why is this 10-year reference period chosen and why not for instance a longer time period (e.g. 15 to 20 year period) or another 10-year period? Is this mainly related to the fact that glaciers have changed a lot, or is this maybe related to the sample size when going further back in time?

Response: Both! We thus clarified the text as follows: "Here, the calibration period was set to the last decade of available reference data as these years best reflect the current mass-change conditions and provide largest glaciological sample size."

§ p.3, l.22: a follow-up question: what influence does the choice of the reference time period have? Would for instance be interesting to see how the sea level contributions for 2016/17 and 2017/18 are affected by the choice of the reference period (i.e. get insight in the sensitivity of your results to the reference time period choice).

Response: In general, the anomaly approach is not sensitive to the choice of the reference period. However, major adjustments of the reference period result in changes of the available glaciological sample (i.e. decreasing back in time) and can force the use of data from glaciers of neighbouring regions, which will influence the results. We added a corresponding statement to the discussion

section 3.2 and we now show the corresponding effect by adjusting the reference in Fig. S3a-e (see also general author's response): "The use of Zemp et al. (2019) as reference dataset has the advantage of analysing the performance of the *ad hoc* estimation at annual time resolution back to the 1960s and allows for adjusting the reference period. Comparing Fig. 1with Fig. S3, we show that the *ad hoc* estimate can be sensitive to the choice of the reference period (i.e., 2006/07–2016/17 in Fig. 1 and 2003/04–2008/09 in Fig. S3a–e), especially when it results in major changes in the glaciological sample such as the use of glaciers from neighbouring regions. As an example, the *ad hoc* estimates for Arctic Canada North change from 7.7 Gt and –93 Gt (Fig. 1c) to 5.5 Gt and –69 Gt (Fig. S3a) for 2016/17 and 2017/18, respectively."

§ p.3, l.25: equation 2: are Bglac,Y,g and Bglac,g,i defined?

Response: We added corresponding definitions to the text and simplified Eq. 2: "For each glacier $g$ with observations in $Y$, we calculated the centred glaciological balance $\beta$ similar to Vincent et al. (2017) as the anomaly of the glaciological balance of the *ad hoc* year $B_{\mathrm{glac},Y,g}$ with respect to the arithmetic mean balance over the calibration period from 2006/07 to 2015/16 $\bar{B}_{glac,2007-2016,g}$:

$$\beta_{Y,g} = B_{\mathrm{glac},Y,g} - \bar{B}_{glac,2007-2016,g} \qquad \text{. Eq. 2"}$$

§ p.3, l.26: 'results show that equation=0': formulation sounds a bit strange to me, as I suppose this is a direct consequence of how it is defined (and not really a result). Consider reformulating this: 'Over the calibration period equation=0' (i.e. omit the 'results show that')

Response: Done.

§ p.3, l.31: 'G': G corresponds to number of glaciers?

Response: Yes, clarified in text.

§ p.4, l.1: equation 3. can you explain why it is not weighted by glacier area? Would intuitively expect this, but probably related to misunderstanding from my side. May be good to shortly explain in text also.

Response: In contrast to the regional mass change (in Gt), we do not expect the regional anomaly of the specific mass balance (in m w.e.) to be dependent on glacier size. To avoid misunderstanding and to improve the clarity of paper, we introduce a short paragraph summarizing the basic assumptions of our approach in section 2.4 and added an additional figure (Fig. S1a-e):

"A change in climatic factors is reflected in a corresponding change of the (regional climatic) Equilibrium Line Altitude (ELA, cf. Cogley et al., 2011) which shifts the vertical mass-balance profile (Fig. S1a–c). Due to different hypsometries, the glaciers of a region can react with a large range of specific mass balances to such a change (Fig. S1d; Kuhn et al. 1985). At the same time, these glaciers are expected to feature common mass-balance anomalies (Fig. S1d; Vincent et al. 2017), i.e., positive or negative deviations for a decrease or increase of the ELA, respectively. Building on these basic assumptions, we calculated the annual *ad hoc* estimate for regional mass changes (Fig. S1e) and corresponding sea-level equivalents for a given ad hoc year of observations Y (e.g. 2017/18) in the following five steps."

§ p.4, l.14: equation 5: where is this 'regional bias of the glaciological sample' used later on?

Response: The regional bias is not directly used for the *ad hoc* estimate but is an important indicator of the representativeness of the glaciological sample for the regional mass change as discussed in Section 3.3., first paragraph.

§ p.4, l.16: 'regional glacier area S': may be good to specify that this is the value for that particular year. This becomes somewhat clear later in the sentence, but nevertheless good to stress this to avoid any possible confusion.

Response: Done: "… by the regional glacier area for that particular year $S_{Y,r}$ (in km$^2$), …"

§ p.4, l.26-27: you explain that reference glacier from neighbouring region is used when no glaciological observations are available for a given region. Little data, so probably not many good options, but this is nevertheless a rough approximation, especially given the distance between the region.

Response: We agree and, hence, conclude that "we need to extend the glaciological sample into so far underrepresented and strongly glacierized regions […]". Of course, numerical modelling and other remotes sensing techniques (e.g. end-of-summer-snowline observations) are possibilities to gain further insights but this goes beyond the scope of the present study.

Three questions here:

o How do you define which one is the neighbouring region when there are several options?

Response: We clarified the text as follows: "For regions with no glaciological observations in the *ad hoc* year, we used available data from neighbouring regions. In line with Zemp et al. (2019), we selected WGMS *reference* glaciers with long-term data series from neighbouring regions that feature a similar mass-balance variability based on qualitative and quantitative criteria, such as a good correlation between mass-balance series available from earlier years (see Table S1)."

o How large is the effect of choosing glaciers from another region: could you quantify this? e.g. with a kind of 'leave-one-out cross validation'?

Response: Such an exercise would be possible but could not solve the main issue with regard to the lack of data: for some regions we needed to use data series that – most probably – do not well reflect the temporal variability of the large ice masses such as in the case of the Patagonian Ice Fields where we had to use mass-balance series from Echaurren Norte to estimate the temporal variability. The suggested 'leave-one-out cross validation' would miss this problem and goes beyond the scope and space of a brief communication. We are happy to investigate this suggestion in a further study.

o Is there no better criterion than proximity of the regions to fill the gap? i.e. would it for instance make sense to determine which regions correlate best for a given reference period in the past (which may not always be the neighbouring I guess) and use this information to fill up the gaps?

Response: In fact, we did consider both the spatial proximity and the correlation of mass-balance series (see response above). In theory, one could use a variance decomposition model as applied in Zemp et al. (2019) but applied to the global dataset. However, this would strongly complicate the interpretation and increases the analysis efforts which both is against the principal aim of our *ad hoc* estimate. It would also go beyond the scope of our study and the limited space of a brief communication.

§ p.5, l.2: 'For each region, we calculated the…': here, or in Zemp et al. (2019)?

Response: We clarified the text as follows: "We combined these overall error bars from Zemp et al. (2019) with an additional uncertainty related to the estimation of the mass-balance anomaly. For the latter, we estimated the uncertainty as 1.96 times the (sample) standard deviation of the mean

§ p.5, l.2-3: '1.96 time the (sample) standard…': why 1.96?

Response: We clarified the text (see above): ", which corresponds to a 95% confidence interval."

**Results and discussion**

§ Global average specific mass change of -0.5 and -1.0 m w.e. yr-1 for 2016/17 and 2017/18 respectively. Uncertainty on these values? Same relative uncertainty as for the global mass change in Gt yr-1 and the sea-level contribution?

Response: We added the uncertainties in the text. They feature the same relative difference between the two years with a larger uncertainty in 2016/17 (in spite of the larger glaciological sample) due to the exclusion of Antarctica (which brings in a large error bar). This issue became obsolete with the new data release including glaciological observations for both 2016/17 and 2017/18.

§ p.5, l.18-25:

o Only little data available for 2017/18. Is well explained why that is, but a pity that is a somewhat incomplete dataset. Would be nice if this could be updated when finalising this manuscript, with the latest data included. Numbers will be used (potentially copied without context?) by other scientists and would therefore be good if these are (close to) the final numbers (i.e. with more glaciers).

Response: We updated the paper and used a new version of the glaciological dataset, recalculated the *ad hoc* estimates, and updated all Tables and Figures.

o It is interesting to see how the method works with different sample sizes, but from the setup used so far it is difficult to estimate what the influence is on the regional mass balances, the global sea level contribution and its uncertainty. Would be very useful if some experiments would be performed to quantify this: e.g. taking only 20-40-60-80- .glaciers and to see where is it crucial to have data: i.e. which regions influence the final numbers the most and where it is problematic (large increase in uncertainty) when data lacks?

Response: In fact, we do show the effect of a reduced sample size at the example of the 40 WGMS *reference* glaciers (see Figs. 2 and S2). A more detailed analysis (20-40-60-80% subsamples) at regional level is not really feasible in view of the overall sample size (i.e., on average only five glaciers per region, when excluding CEU) and goes beyond the scope of the paper and the space of a brief communication.

§ p.5., l.11 & l.21: uncertainty on the global mass change is substantially lower for 2017/18 (138 Gt yr-1) compared to 2016/17 (249 Gt yr-1). Why is this? Would intuitively expect the opposite, given the limited number of glaciers considered in 2017/18 (70) vs. 2016/17 (150). This comment is related to previous suggestions to quantify the effect of glacier sample size and the effect of the location/region on the uncertainty.

Response: The larger uncertainty in 2016/17 (in spite of the larger glaciological sample) is due to the exclusion of Antarctica (which brings in a large error bar). This issue became obsolete with the new data release including glaciological observations for both 2016/17 and 2017/18.

§ p.5, l.21: mass loss of 512 Gt yr-1: would reformulate to mass change of -512 Gt yr-1 to be consistent with formulation for 2016/17 and avoid any confusion. My first reaction when

reading this was: 'oh, a positive mass balance?' and then I re-read it and saw that here you refer to a loss vs. a mass change.

Response: Done.

§ p.5, l.25: '…contributor**s**.'

Response: Done.

§ p.6, l.9: 'again relative good agreements again in the…' (omit one of the two again's)

Response: Done.

§ p.6, l.14-15: 'in years with small data samples and strong anomalies it remains arguable which of the two approaches better represents…': would it be feasible to quantify this? Would be interesting addition.

Response: In the discussion (Section 3.2), we state that "[…], the variance decomposition model as used by Zemp et al. (2019) tends to reduce the variance for statistically small samples since it only extracts the common year-to-year variability found in all glaciological time series of a region. The variability (at each glacier) that is not found at other locations is assigned to the residual (i.e. unexplained variance). Therefore, our ad hoc estimate is generally well suited to assess the global value of the more representative reference data. However, in years with small data samples and strong anomalies it remains arguable which of the two approaches better represents the true global glacier mass changes."

Hence, we think that the variance decomposition model has the tendency to reduce the variance too much in years with small samples and extreme mass changes. A quantification would be possible through detailed regional analysis and comparisons to independent datasets. However, this remains to be addressed in future studies.

§ p.6, l.28: at the end of this paragraph: would expect that you explain what the consequence of this statement is: e.g. "This does imply that…"

Response: We added the following sentence: "This implies that our approach allows for a regional selection of reference datasets and, hence, can be used in future consensus estimates of global glacier mass changes."

§ p.7, l.7: 'regional biases range between -0.6 and +0.5 m w.e. yr-1': this is quite large, in the order of the signal almost, no?

Response: Yes. We clarified the text as follows: "At regional level, the bias ranges between −0.6 and +0.5 m w.e. yr$^{-1}$. This confirms that the glaciological observations are well suited to cover the temporal variability but not necessarily the absolute value of regional glacier mass changes. At global level, the bias averaged out for the area-weighted mean. However, this is rather fortuitous and can change with the use of a different reference dataset (e.g., Gardner et al., 2013; Wouters et al., 2019)."

§ p.7, l.5-16: ok, interesting! But what if you work with a method more sophisticated than taking values from neighbouring regions? (see earlier comment on this)

Response: See response to comments related to § p.4, l.26-27 (above).

**Conclusion**

§ Explains nicely why this work is important and why we need such updates. When you present contributions to sea level (p.7, l.23-24), could you compare these numbers to the contribution from the ice sheets and thermal expansion? (for these 2016/17 and 2017/18 specifically, and if not available, with the numbers from the last years/decade) Would be good to stress, once again, the important sea-level contribution from glaciers, which ice sheet modellers quite often tend to forget/underestimate.

Response: Done: "… and resulted in annual global glacier contributions to sea-level rise exceeding 1 mm SLE per year, which corresponds to more than a quarter of the observed sea-level rise (cf. IPCC, 2019)."

§ p.7, l.25-28: importance of glaciological sample is stressed. Fully agree and think that this study nicely supports this statement. It would however also be justified to put this a bit in perspective and explain that in the future we will be able to rely more and more on remote sensing observations. These observations will become available at higher resolution, with smaller uncertainties and biases, and most importantly: at a high temporal resolution (with sub-annual update). As such, they could therefore be used in addition to glaciological measurements (still a long way to go for replacing them…), making us less dependent on direct field measurements.

Response: In-situ measurements will always be needed for process understanding as well as for calibration and validation of remote sensing data and numerical modelling studies (e.g. snow/firn density). We fully agree on the large potential of remote sensing data for absolute volume change estimates over entire mountain ranges. However, density conversion issues will limit the possibility of deriving annual or seasonal mass change variability from geodetic surveys (including altimetry sensors such as ICESat-2).

In the conclusions, we strengthened the statement about the remote sensing data: "and we need to tap the full potential of space-borne surveys to further improve the spatio-temporal coverage and resolution of the reference dataset."

REVIEWER #2

This is a nice, simple, and clear paper…

Response: We thank the reviewer for the positive evaluation and the constructive feedback.

…, but I am not convinced about the uncertainty budget (p5 line1-5). This needs to be fixed or argued better. You are assuming that there is a very simple relationship between βavg and B. Basically a line with slope of 1 and an intercept that you estimate. I would like to see supplementary scatterplots of βavg vs B, with that line you estimate. One plot per region, with one point for every calibration year. You could also add the two ad-hoc estimates with error bars. This would be valuable because it instinctively gives you an idea if the slope=1 is a reasonable assumption, how large the scatter is, and give you an idea if you trust the ad-hoc estimates.

Response: We addressed this feedback by (i) adding a section (2.4) and figure (Fig. S1a-e) summarizing the basic assumptions of our approach, (ii) clarifying the section about the uncertainty estimates, (iii) adjusting and better explaining Eq. 4, and (iv) adding additional scatterplots (Fig. S2a-s) and corresponding results (Table S2) to compare our approach (slope = 1) with version where the slope is derived from the linear regression between specific mass changes of the reference data and centred glaciological mass balances over the reference period from 2006/07 to 2015/16.

(i) Section 2.4 explaining our basic assumptions: "A change in climatic factors is reflected in a corresponding change of the (regional climatic) Equilibrium Line Altitude (ELA, cf. Cogley et al., 2011) which shifts the vertical mass-balance profile (Fig. S1a–c). Due to different hypsometries, the glaciers of a region can react with a large range of specific mass balances to such a change (Fig. S1d; Kuhn et al. 1985). At the same time, these glaciers are expected to feature common mass-balance anomalies (Fig. S1d; Vincent et al. 2017), i.e., positive or negative deviations for a decrease or increase of the ELA, respectively. Building on these basic assumptions, we calculated the annual *ad hoc* estimate for regional mass changes (Fig. S1e) and corresponding sea-level equivalents for a given *ad hoc* year of observations Y (e.g. 2017/18) in the following five steps."

(ii) Page 5, lines 1ff, clarifying uncertainty estimates: "We combined these overall error bars from Zemp et al. (2019) with an additional uncertainty related to the estimation of the mass–balance anomaly. For the latter, we estimated the uncertainty as 1.96 times the (sample) standard deviation of the mean centred glaciological balance for each region over the calibration period from 2006/07 to 2015/16 (cf. Eq. 3), which corresponds to 95% confidence intervals. In cases with only one glacier in the glaciological sample (resulting in a standard deviation of zero), we set the uncertainty to 100% of the anomaly. The two errors related to the reference dataset and to the mass–balance anomaly were combined according to the law of random error propagation."

(iii): Adjusted and better explained Equation 4:

$$B_{\text{adhoc}.Y,r} = m \cdot \bar{\beta}_{Y,r} + \bar{B}_{\text{ref},2007-2016,r} \; . \qquad\qquad (4)$$

"Basically, this corresponds to a linear regression model with slope $m$ and y-intercept $\bar{B}_{ref}$ (Fig. S2). We set $m$ = 1 in order to make the *ad hoc* estimation applicable to reference data without annual resolution (cf. Fig. S3, Table S2). "

(iv): See revised paper for additional adding additional scatterplots (Fig. S2a-s) and corresponding results (Table S2).

Paper structure, abstract, and conclusion: good.

Response: Thanks.

Minor comments:

The equations are so simple that they are almost unnecessary. I would rather have an illustrative visualization explaining the method (one of the scatter plots).

Response: We agree that the equations are simple but prefer keeping them for the reason of clarity and traceability.

p1 line 16: Why not the estimates to the abstract.

Response: We added the values and error bars to the abstract.

p3 line 30: Here you essentially assume that the arithmetic average of all the β's is representative of the glaciers throughout the region. However, I imagine that β will correlate with e.g. glacier size, and that the sample is not representative in terms of size (and other characteristics). I don't expect this to introduce a major bias, but the possibility should probably be considered when estimating the errors. I propose that you explicitly mention the assumptions at this point.

Response: In contrast to the regional mass change (in Gt), we do not expect the regional anomaly of the specific mass balance (in m w.e.) to be dependent on glacier size. In order to avoid misunderstanding and to improve the clarity of paper, we introduce a short section summarizing the basic assumptions of our approach in section 2.4 (see above, response to Reviewer 1 on p. 4, l. 1, Eq. 3).

equation 2 and 3 (and...): It is good to be clear, but many of these equations are perhaps a bit over-verbose. These two eqns. basically just define what a mean value is.

Response: We simplified Eq. 2 but decided to keep Eq. 3 for clearly showing the difference between $\beta_{Y,g}$ and $\beta avg_{Y,r}$.

p4 line 10: Is it a stable glacier sample? What do you mean by stable?

Response: We clarified the text: "stable glacier sample (i.e., observations available from the same glaciers in all years), …". For several regions, the glacier sample is not stable, which means that for some (or all) glaciers with observations in the *ad hoc* year, there are missing data for some (or all) years of the reference period. This problem increases when moving the reference period back in time. In fact, this issue with variable glacier samples is why we use the anomaly approach for our *ad hoc* estimation: using a simple bias correction (Eq. 5) for an *ad hoc* estimation would be much more sensitive to changes in the glacier sample since it uses absolute mass-balance values instead of anomalies (see above, new paragraph in section 2.4 on basic assumptions).

p4 lines 15-20 and equation 6: I assume that the 1/1000 is a unit conversion. Please remove this. The method is independent of units and we trust you to do the conversions correctly when you report the results.

Response: Yes, this is a unit conversion. We removed it in the Eq. 6 together with the corresponding unit in the text.

p4 equation 7: Please remove the 10^6 unit-conversion multiplier.

Response: Yes, this is a unit conversion. We removed it in the Eq. 7 together with the corresponding unit in the text.

p4 line 25: This is on the other hand important to specify.

Response: Agreed and not changed.

p4 line 27: There is only one region with n.a. in table 1 — Please specify which region at this point.

Response: No, there are several regions without glaciological observations as indicated in Table S1. (In contrast, Table 1 of the original manuscripts refers to the available *ad hoc* estimates – with one region (i.e. ANT) without result. In the revised version, Table 1 has data for all regions.)

p5 line 1-5: This method seems rather arbitrary. Is it really mathematically justified? E.g. this method has no consideration of how representative the sample is. Consider also the case where the sample is complete (has the entire population). In this case you this error term should be zero (right?). What would your current 1.96x std method give?

Response: We clarified the text: "For the latter [mass–balance anomaly], we estimated the uncertainty as 1.96 times the (sample) standard deviation of the mean centred glaciological balance for each region over the calibration period from 2006/07 to 2015/16 (cf. Eq. 3), which corresponds to 95% confidence intervals." Note that this error term provides a measure of the variability of the

mass-balance anomaly in a region. It becomes zero in the ideal case of our basic assumption (see response above) that all glaciers in a region react with a common anomaly to a climatic change. In our data sample, we will not have a perfectly common anomaly for all glaciers and, hence, do not expect the corresponding variability to become zero (even not for the full population). The combination of different (independent) error sources according to the law of random error propagation is mathematically well justified.

Would it not be better to estimate the uncertainty from the residuals. e.g. using a jack-knife approach. You could withhold the data for one of the years in the 2007-2016 and see how well your method can predict the with held data. That would give you 10 residuals for each region. From these you can estimate both an RMS error and a mean bias. It would even work if your sample has only a single glacier.

Response: In fact we do a similar test (using the years before the reference periods) and show that the $r^2$ decreases from 0.65 for the full sample to 0.38 for the reduced sample size (cf. Fig. S4). Using jack-knife estimates would be possible but would come with the major shortcoming that the approach would be restricted to reference datasets with annual time resolutions (e.g. Zemp et al. 2019) but not for others currently just providing decadal change rates (Bolch et al. 2013, Gardner et al. 2013,  Wouters et al. 2019; cf. Fig. S3).

p7 line 13-14: I would add "... using this method." to this sentence. Maybe another statistical treatment than your proposed method would have better performance.

Response: Done: "This low performance suggests that – for the present approach – the WGMS reference glacier sample alone is too small and represents too few regions for an *ad hoc* estimation of global glacier contributions to sea-level rise."

equation 6: Does it have any practical effect to allow for changing area rather than just using a constant? It would make it even easier to get ad-hoc estimates, if you did not need new estimates of S.

Response: It makes sense to account for glacier area changes since the differences in time (and area) between the survey year of the RGI and the *ad hoc* year can be substantial. As such, the glacier inventory of New Zealand in RGI 6.0 dates from the 1970s. Most other regions have survey years from 10 to 20 years back in time with relative area change rates of up to 1% per year.

p3 line 27: Is the sentence starting with "Over the calibration period ..." necessary? Isnt it clear it must be so, given the definition?

Response: Removed.

p2 line 4: "))"

Response: Deleted.

p1 line 27: "AR 5"->"AR5"

Response: Corrected.

p6 line 9: There is a limit to how far you can extrapolate the empirical calibration. The proposed scatterplots would help diagnosing the problem.

Response: We agree and clarified the text as follows: "Here, the calibration period was set to the last decade of available reference data as these years best reflect the current mass-change conditions and provide the largest glaciological sample size."

Figure S2: Please expand on the description in the caption and page 6 line24-26. It is not sufficient to understand what has been done.

Response: Done.

---

## Author Response (AR1)

[revised manuscript text omitted]

**Table S2** *Ad hoc* estimates of regional mass changes for 2016/17 and 2017/18 according to different regression models. Specific mass changes ($B_{adhoc}$) are shown for *ad hoc* estimates (Eq. 4) with slope *m* equal to 1 (as used in Table 1) and with *m* as derived from linear regressions (as shown in Fig. S2), together with corresponding differences ($\Delta B$) for both years. Large values for $\Delta B$ are found in regions with small slopes and low correlation coefficients (e.g. New Zealand).

| Region | *m* = 1 | | *m* from regression | | $\Delta$ | |
|---|---|---|---|---|---|---|
| | $B_{adhoc}$ 2017 (m w.e.) | $B_{adhoc}$ 2018 (m w.e.) | $B_{adhoc}$ 2017 (m w.e.) | $B_{adhoc}$ 2018 (m w.e.) | $\Delta B$ 2017 (m w.e.) | $\Delta B$ 2018 (m w.e.) |
| 01 Alaska | −1.37 | −2.29 | −1.24 | −1.92 | −0.13 | −0.37 |
| 02 Western Canada & USA | −0.68 | −0.85 | −0.78 | −0.84 | 0.10 | −0.01 |
| 03 Arctic Canada North | 0.07 | −0.90 | −0.07 | −0.82 | 0.14 | −0.08 |
| 04 Arctic Canada South | −0.22 | −0.90 | −0.28 | −0.84 | 0.06 | −0.06 |
| 05 Greenland | −0.27 | −0.44 | −0.49 | −0.55 | 0.22 | 0.11 |
| 06 Iceland | −0.11 | 0.14 | −0.17 | 0.03 | 0.06 | 0.11 |
| 07 Svalbard & Jan Mayen | −0.57 | −0.69 | −0.58 | −0.71 | 0.01 | 0.02 |
| 08 Scandinavia | −0.09 | −1.48 | −0.09 | −1.46 | 0.00 | −0.02 |
| 09 Russian Arctic | −0.69 | −0.80 | −0.65 | −0.74 | −0.04 | −0.06 |
| 10 North Asia | −0.67 | −0.17 | −0.52 | −0.30 | −0.15 | 0.13 |
| 11 Central Europe | −1.60 | −1.43 | −1.59 | −1.42 | −0.01 | −0.01 |
| 12 Caucasus & Middle East | −0.89 | −0.28 | −0.90 | −0.45 | 0.01 | 0.17 |
| 13 Central Asia | −0.39 | −0.11 | −0.25 | −0.13 | −0.14 | 0.02 |
| 14 South Asia West | −0.34 | 0.17 | −0.12 | 0.00 | −0.22 | 0.17 |
| 15 South Asia East | −0.77 | −1.10 | −0.49 | −0.59 | −0.28 | −0.51 |
| 16 Low Latitudes | −1.13 | −0.29 | −1.07 | −0.71 | −0.06 | 0.42 |
| 17 Southern Andes | −0.13 | −1.11 | −0.26 | −1.11 | 0.13 | 0.00 |
| 18 New Zealand | 0.13 | −2.62 | −0.59 | −0.89 | 0.72 | −1.73 |
| 19 Antarctic & Subantarctic | −0.52 | −0.16 | −0.20 | −0.12 | −0.32 | −0.04 |

[Figure]

**Figure S1** *Ad hoc* estimation of regional mass changes exemplified with glaciological data from Hintereisferner (HEF), Vernagtferner (VGT), and Kesselwandferner (KWF) located in the Ötztal, Austria. **(a–c)** Vertical mass-balance profiles (B, lower horizontal axis) for the reference period from 2006/07 to 2015/16 (light blue) are plotted together with the corresponding arithmetic average (dark blue) for the three glaciers. Glacier hypsometries are shown as horizontal grey bars (S, upper horizontal axis). **(d)** Boxplots showing mean (green triangle), median (orange line), and distribution of annual means (avg) and annual standard deviations (stdv) of both mass balance (B) and mass-balance anomaly ($\beta$, cf. Eq. 3) of the glaciological sample (i.e., HEF, VGT, KWF) over the observation period (i.e., from 2006/07 to 2015/16). On average, the annual mass balance of the three glaciers was –0.82 m w.e. and $\beta$ – by definition – is zero over the reference period. Note that the mean annual standard deviation of the mass-balance anomalies ($\beta_{stdv}$) is typically smaller than the one of the mass balance ($B_{stdv}$; here by about three times). For this reason, we expect the anomaly approach to perform better than a simple bias correction. **(e)** Plots of glacier mass balance versus mass-balance anomaly ($\beta_{avg}$) for the glaciological sample (green circles), the reference data by Zemp et al. (2019, blue squares), and the *ad hoc* estimate (orange circles). The latter (orange, cf. Eq. 4) is basically obtained by vertically shifting the regression of the glaciological sample (green line with slope $m = 1$ and intercept b = –0.82 m w.e.) to fit the intercept (b = –0.87 m w.e.) of the reference data regression (blue line with statistics given at the bottom of plot).

[Figure]

**Figure S2** Relationship between annual specific mass changes of the reference data and glaciological mass-balance anomalies over the reference period from 2006/07 to 2015/16. For all regions (a–s), linear regressions (blue line) are plotted with corresponding statistics (slope $m$, intercept $b$, coefficient of determination $r^2$, and root mean square error $rmse$) at the right bottom of the plots. *Ad hoc* estimates for 2016/17 and 2017/18 are shown for $m = 1$ (orange dots) – as used in Fig. 1

10   and Table 1 – and for $m$ as derived from the regression (red crosses); corresponding values are compared in Table S2. Plots are ordered from top left to bottom right according to the region numbers in RGI 6.0 (see Table 1).

[Figure]

**Figure S3** *Ad hoc* estimates of selected regional mass changes in 2016/17 and 2017/18 based on different reference datasets. The plots (**a–e**) in the left column correspond to plots in Fig. 1 using Zemp et al. (2019) as reference dataset but for the reference period 2003/04–2008/09. The plots in the middle column (**f–j**) use Gardner et al. (2013, 2003/04–2008/09) as reference dataset. The plots in the right column (**k–n**) are using Wouters et al. (2019, 2005/06–2014/15) or Bolch et al. (2013, 2003/04–2007/08) as reference dataset. Note that within a region, the annual anomalies (pale blue and pale red) are similar but absolute mass changes (in Gt) vary strongly in case of different mass-change rates in the reference datasets. For Greenland (**d, i, n**), no *ad hoc* estimate were calculated for 2017/18 because the glaciological observations are from a glacier without data in the reference periods.

[Figure]

**Figure** S4 Annual global mass changes in comparison between the *ad hoc* estimates of this study and the reference dataset by Zemp et al. (2019). The comparison is shown for the *ad hoc* estimates as based on the full glaciological sample of corresponding years (**a**) and for the WGMS *reference* glaciers only (**b**). The linear regression refers to the fit between the values (green) over the validation period (2006/07−2015/16, cf. Fig. 2).

| Page 6: [1] Formatted | mzemp | 14.01.2020 08:58:00 |
|---|---|---|

Highlight

| Page 6: [2] Formatted | mzemp | 14.01.2020 09:00:00 |
|---|---|---|

Highlight

| Page 6: [3] Formatted | mzemp | 14.01.2020 09:03:00 |
|---|---|---|

Font: Italic